# The Effect of Combining Post-Harvest Calcium Nanoparticles with a Salicylic Acid Treatment on Cucumber Tissue Breakdown via Enzyme Activity during Shelf Life

**DOI:** 10.3390/molecules27123687

**Published:** 2022-06-08

**Authors:** Mohamed F. M. Abdelkader, Mohamed H. Mahmoud, Lo’ay A. A., Mohamed A. Abdein, Khaled Metwally, Shinya Ikeno, Samar M. A. Doklega

**Affiliations:** 1Department of Plant Production, College of Food and Agriculture, King Saud University, Riyadh 12372, Saudi Arabia; mohabdelkader@ksu.edu.sa; 2Department of Biochemistry, College of Science, King Saud University, Riyadh 12372, Saudi Arabia; mmahmoud2@ksu.edu.sa; 3Pomology Department, Faculty of Agriculture, Mansoura University, Mansoura 35516, Egypt; 4Seed Development Department, Agricultural Professions Syndicate, Downtown, Cairo 11669, Egypt; 5Department of Genetics, Faculty of Agriculture, Ain Shams University, Cairo 11241, Egypt; khaleda.fatah@agr.asu.edu.eg or; 6Department of Biological Functions Engineering, Graduate School of Life Science and Systems Engineering, Kyushu Institute of Technology, 2-4 Hibikino, Wakamatsu, Kitakyushu 808-0196, Japan; 7Floriculture and vegetable Department, Faculty of Agriculture, Mansoura University, Mansoura 35516, Egypt; samar_2005@mans.edu.eg

**Keywords:** shelf life, cucumber, cell wall degradation enzymes, nano calcium particles and salicylic acid

## Abstract

In the present study, an experiment was carried out on the postharvest of cucumber fruit during a 14-day shelf life. The aim was to assess the impact of calcium nanoparticles (CaNPs) blended with different concentrations of salicylic acid (SA) on the shelf life of cucumbers during the seasons of 2018 and 2019. The investigation further monitored the influences of CaNPs-SA on some physical properties of cucumber, including the percentage weight loss, color, and fruit firmness. In addition, chemical properties, such as total soluble solids (SSC%), total acidity (TA%), total soluble sugars, and chlorophyll pigmentation of the fruit skin, were assessed during a 14-day shelf lifeCell wall degradation enzymes (CWEAs) such as polygalacturonase (PG), cel-lulase (CEL), xylanase (XYL), and pectinase (PT) were also researched. In addition, the generation rates of H_2_O_2_ and O_2_^•−^ were calculated, as well as the reduction of DPPH. The lipid peroxidation (malondialdehyde, MDA) and cell membrane permeability (IL%) of cell wall composites were also determined. CaNPs-SA at 2 mM suppressed CWEAs, preserved fruit quality, reduced weight loss throughout the shelf-life period, and reduced the percent leakage value. At this concentration, we also found the lowest levels of MDA and the highest levels of DPPH.

## 1. Introduction

In Egypt, cucumber (*Cucumber sativa* L. cv Barracuda) is the most popular vegetable crop grown for local and global consumption. Additionally, it is a preferred product around the world Cucumber growth generally increases throughout the summer season; in Egypt, a total area of 226,385 ha is used to grow cucumber, yielding approximately 364,571 tons [1]. Cucumber quality decreases significantly after harvesting due to a loss of water, shriveling, and yellowing from loosening skin chlorophyll pigment; as a result, cucumbers have a shorter shelf life in the market, lasting about 2–3 days [2]. Cucumber is a non-climacteric vegetable crop that contains more than 90% water. The primary problem that influences fruit quality during processing is often excessive moisture loss [3]. The outcomes of weight loss result in a lower marketable income. In addition, fruits are more sensitive to infection by postharvest pathogens [4]. Naturally, fruits and vegetables have many layers of wax on their surfaces [5]. The wax diminishes the rate of water evaporation in fruit tissues during the growth, development, or post-harvest stages [6]. These layers vary in thickness throughout the circulation stages and are affected by processing stages, such as washing, where they are easy to remove [7]. Moreover, many variables promote the deterioration and waste of fruits during storage, including environmental factors [8], harvest time, the stage of fruit maturity, and the occurrence of mechanical damage throughout handling phases [9]. Previously, coating techniques have been implemented to improve the shelf life and quality of fruits and vegetables [10], or to develop deliberate cucumber characteristics throughout the selling chain for consumers [11]. However, deciding on the right coating blend assuredly affects the performance of other coating layers. Choosing a film-type coating for fruit is typically a safe and effective way to preserve cucumber quality by reducing fruit rotting and minimizing weight loss. [12].

The recent application of advanced techniques, such as nanotechnology, in the post-harvest of fruits and vegetables merits further investigation. The nano-technique, which works to improve a material’s physical and chemical properties, also has significant antifungal and antiviral properties. [13]. It also has uses in different fields, such as medicine [14] and pharmacology [15]. Furthermore, in agriculture for horticultural products, nanotechnology has a positive effect on the preservation of fruits and vegetables during storage, thereby increasing their shelf life [16].

Salicylic acid (SA) is recognized as a phenolic compound that has effects on plant growth and defenses against different stresses that occur simultaneously during the postharvest of fruit [17]. Fruit tissues (cells) produce reactive oxygen species (ROS) when exposed to stress conditions [18]. ROS include superoxide (^1^O_2_), peroxide (H_2_O_2_), and hydroxyl (OH^•^) ions, which cause damage under stress conditions [19]. As a result of stress damage in cellular structures, ROS are also produced. [20]. In addition, SA controls multiple physiological and biochemical paths in cells [21]. It inhibits the harmful effects of these ROS by enhancing antioxidant activities, such as lowering H2O2 levels through the action of ascorbate (APX) [22]. Additionally, SA restrains fruit senescence throughout the duration of storage [23,24], maintains elements of fruit quality [25], reduces fungal rotting infection in susceptible vegetables [13,26], and efficiently influences resistance to chilling [27].

Therefore, in this investigation, cucumber ‘Barracuda’ fruits were treated with CaNPs blended with salicylic acid at varying doses and then stored at the ambient condition. Enzymatic activity was then measured to see how the CaNPs-SA blends affected tissue breakdown over the course of the shelf-life.

## 2. Results

### 2.1. Physical Properties

Figure 1 illustrates the variations in the physical properties of cucumber fruit (weight loss percentage, fruit color hue angle, and firmness) during shelf life. There was a significant impact of CaNPs-SA applications on cucumber fruits at *p* < 0.05 when considering storage duration (days) as a factor. It can be seen from the figure that the variations between CaNPs-SA treatments for the weight loss percentage of cucumber fruits were remarkable on the fourth day of shelf life. The weight loss percentage increased gradually overall among all CaNPs-SA treatments during shelf durations. Cucumber fruits treated with 2 mmol L^−1^ CaNPs-SA had a significantly lower weight loss percentage (19.74%) on the 14th day of storage. However, at the end of the experiment, other CaNPs-SA treatments independently showed a higher weight loss percentage at the same recorded interval—control (37.35 percent), CaNPs-SA 0 mM (35.14 percent), and CaNPs-SA 1 mM (30.25 percent). The fruit color hue angle (h°) was also monitored during the 14-day shelf life. Cucumber color (h°) decreased independently according to CaNPs-SA treatments during 14 days of storage. The fruits treated with CaNPs-SA in 2 mmol L^−1^ treatments showed a high hue angle throughout the storage duration. In addition, cucumber fruit firmness (Figure 1) showed an interaction between CaNPs-SA treatment and shelf-life duration (days) at *p* < 0.001 when both were admitted as experimental factors. Fruit firmness was higher at harvest time and decreased continually and gradually up until the end of the shelf-life period. It was noted that the treatment of cucumber fruits with CaNPs-SA at 2 mmol L^−1^ better maintained fruit firmness compared to the other treatments during the storage period.

### 2.2. SSC%, TA%, and SSC/TA Ratio

Figure 2 shows a significant interaction at *p* ≤ 0.001 between shelf-life duration in days and CaNPs-SA application. In comparison to the initial value at harvest time, the total soluble solid content (SSC percent) grew modestly and steadily throughout the CaNPs-SA treatments up to the end of the shelf-life period. However, TA percent declined during storage, with larger declines than the original levels at harvest. When compared to control fruits and fruits treated with CaNPs-SA during shelf life, the coated CaNPs-SA treatment at 2 mM provided greater stability in these parameters. On the 14th day of shelf life, the SSC percentage was 2.73 percent, the TA percentage was 0.062 percent, and the total sugar percentage was 0.54 percent, compared to the control fruits, which had an SSC percentage of 3.92 percent, a TA percentage of 0.0.039 percent, and a total sugar percentage of 0.62 percent.

### 2.3. Cucumber Chlorophyll Pigment Content

Figure 3 shows the variations in fruit skin pigment (chlorophyll content) as a function of shelf-life duration in days. It can be seen from the figure that chlorophyll pigments presented a significant interaction between variables at *p* ≤ 0.001 when the shelf-life duration (days) and CaNPs-SA applications were analyzed. Observably, all chlorophyll elements progressively decreased throughout the storage phase. The decreases were caused by a decrease in the salicylic acid content of the CaNPs. The fruits that were immersed in 2 mM CaNPs-SA treatments presented with greater preservation of chlorophyll compartments up to the end of the experiment period. The results were recorded (*Chl_a_* = 6.79, *Chl_b_* = 3.04, and *Chl _a+b_* = 9.83 mg g^−1^ FW) compared to the initial value at harvest time and other treatments. The control cucumber treatment showed a more rapid degradation in chlorophyll pigment at the same interval (*Chl_a_* = 5.84, *Chl_b_* = 2.54, and *Chl _a+b_* = 8.18 mg g^−1^ FW).

### 2.4. Effect of CaNPs-SA Treatments on Cell Wall Degradation Enzyme Activities

Figure 4 shows the activities of cell wall-degrading enzymes (CWEAs), such as PG, CEL, XYL, and PT, during the shelf-life period (days) and CaNPs-SA treatments as factors. When time and CaNPs-SA treatments were considered, there was a significant interaction (*p* < 0.001). Initially, CWEAs increased gradually after harvesting and immersion in different CaNPs-SA treatments and continued to increase up to the 14th day of shelf life. The CaNPs-SA 2 mM treatment reduced overall CWEAs compared to the other treatments and control fruit. However, CWEAs increased more quickly in control fruits, and all enzymes increased until the 14th day of shelf life. Notably, the degrading enzyme activities increased independently of the CaNPs-SA treatments during shelf life and decreased with increasing SA concentrations. However, cucumber fruits that were treated with CaNPs-SA at 2 mM presented lower CWEAs during the 14-day shelf-life period.

### 2.5. O_2_^•−^ and H_2_O_2_ Production, DPPH Reduction, LOX, and Percent of Ion Leakage

The changes in O_2_^•−^ and H_2_O_2_ output rate and the decrease in DPPH as a function of shelf-life duration are shown in Figure 5 and Figure 6. When the shelf life (days) and CaNPs-SA treatments were evaluated, the O_2_^•−^ and H_2_O_2_ formation rates showed a significant interaction at *p* < 0.001. During the 14-day shelf-life trial, the output of O_2_• and H_2_O_2_ increased steadily for all applications. Over the course of the storage time, the control fruits produced more O_2_^•−^ and H_2_O_2_ than the other treatments. On the 14th day of storage, however, fruits treated with CaNPs-SA at 2 mM exhibited considerably lower O_2_^•−^ and H_2_O_2_ generation (0.32 and 0.10 nmol min^−1^ g^−1^ FW, respectively) compared to the control fruits (0.42 and 0.18 nmol min^−1^ g^−1^ FW, respectively).

## 3. Discussion

The observed variations in our results may be explained by the high concentration of SA blended with the calcium nanoparticles, which could have inhibited the cell wall degradation enzyme activities (CWEAs) during shelf life [28]. Our results confirmed that the CANPs-SA treatment at 2 mmol L^−1^ inhibited CWEAs considerably more than the other treatments and control fruit throughout storage period (Figure 4). As a result, the fruit firmness was affected. Fruits treated with CaNPs-SA at 2 mM were firmer during the shelf-life period (Figure 2). This could also be due to the increased inhibition of cell wall degradation enzymes, such as PG, CEL, and XYL, all of which are linked to fruit firmness [29]. Because of the presence of salicylic acid, the water content of the cucumber fruit tissue was preserved during the shelf life by inhibiting CWEAs [5,27,30,31].

These outcomes could well be correlated with the impact of treatment with CaNPs loaded with 2 mM salicylic acid. This treatment lowers water transpiration [32], delays fruit ripening/senescence [23], and activates antioxidant enzymes during storage, giving additional protection to the cucumber tissue/cells against oxidative reactions [33]. Antioxidant enzyme system activities are increased when CS/PVP and SA are combined at 2 mM, resulting in decreased lipid peroxidation accumulation [16,34,35,36].

The changes in chlorophyll compartments over time could be attributed to a decrease in antioxidant enzyme activity [37,38]. As a result, more ROS are created, resulting in increased oxidative reactions, such as lipid peroxidation and protein oxidation. Following this, further malfunction develops in cell membrane structures along with an increase in IL percent [39], finally leading to cell death [27,40,41].

As a result, the observed treatment effects were most likely the result of CaNPs and SA limiting cell wall hydrolysis by modulating CEL, PG, and PT. [23]. An improvement in fruit firmness throughout the shelf life was associated with a faster increase in endogenous SA in fruit tissue (Figure 1) [42]. Interestingly, the low level of CWEAs suggests that PG can inhibit SA during storage (Huber, 1983), and the reduction in CEL activity caused by CaNPs-SA at 2 mM could be linked to a structural change in the hemicellulose required for fruit firmness or softening [29]. According to the findings of [43], CaNPs in the presence of SA at 2 mM reduce fruit senescence or ripening by limiting CWEAs and suppressing ethylene production.

Reduced antioxidant enzyme activity could be the cause of the increases in MDA, LOX, and IL percent with control treatments over the shelf-life period [37]. As a result, more ROS are produced, resulting in increased oxidative responses in terms of lipid peroxidation and LOX activation. Following this, there is an increased disruption of the cell membrane structure and higher IL percent [16], followed by cell death [33].

Other metabolic activities, such as aerobic respiration, could be responsible for the high quantities of O_2_^•^ and H_2_O_2_ produced during the experiment [44]. As a result, increasing AEAs, such as SOD, may improve fruit tissue tolerance to O_2_^•−^, and increasing CAT and APX activities could aid in the scavenging of both O_2_^•−^ and H_2_O_2_ during storage [45].

## 4. Materials and Methods

### 4.1. Fruit Materials

Immature cucumber (*Cucumis sativus* L. cv Barracuda) fruits were picked in 2018 and 2019 from a commercial farm in Mansoura, Dakahlia, Egypt. Upon arrival at the Mansoura University Floriculture and Vegetable Department, the fresh fruits were handpicked for uniformity in size and shape and the absence of external injury. The fruits were cleaned with chlorinated water (0.05% NaOCl), flushed with distilled water, and permitted to air dry. A total of 192 fruits were selected for the shelf-life experiment and were allocated toward pair lots; each lot contained 96 fruits. The first lot was used to measure the physical naturalistic variables and the second batch was designated for chemical analysis throughout the 14-day duration of shelf life. The physical and chemical examinations were judged every two days throughout the continuation of the research experiment.

### 4.2. Use of Salicylic Acid in Calcium Carbonate Nanoparticle (CaNP) Production

Calcium nanoparticles (CaNPs) were produced using a modified version of the procedure described in [46]. The CaNPs were designed by introducing SA at 1 and 2 mM with filtered water. The SA was introduced using pure water and CaCl_2_ liquid at 50 mM. The mixture was shaken for one hour at 5000 rpm and then stored at room temperature for three days. The mixture was then shaken for another hour at 5000 rpm before being left at room temperature for three days.

### 4.3. Nanoparticle Characterization Utilizing UV-Vis Spectroscopy

Using the ATI Unicom UV-vis spectroscopic analysis image program (Ver. 3.20), the reduction in absolute Ca^++^ particles and the increase in succeeding calcium nanoparticles were detected by identifying the UV-Vis color curve of the response mixture at multiple wavelengths. The UV-Vis spectra of the mixed CaNPs were observed to be in the 240–440 nm range. This experiment was carried out at 25 °C using quartz cuvettes (1 cm optical way, Figure 7).

### 4.4. CaNP Characterization via Zeta Potential

Zeta potential analysis (ZPA) was used to define the situation of the CaNPs at the start and to check the equilibrium of the CaNP liquid at the end. The Electron Microscope Lab determined the combination of CaNPs, including the SA external charge, using the Malvern Instruments Ltd. Zeta Potential software (Ver. 2.3). The CaNPs-SA blend had an exterior charge that attracted a small layer of differently charged anions to its surface. CaNPs were encased in two layers of anions that moved around as they were dispersed throughout the mixture. The electric potential near the end of the duplicated layer is called the zeta potential, and it varies from +100 mV to −100 mV (Figure 8). After CaNPs were integrated with SA, the ZPA rate was 4.74 mV (highly stable). CaNPs with ZPA rates greater than or equal to +25 mV are usually considered as being more stable [47].

### 4.5. Investigation of CaNP Properties via Transmission Electron Microscopy (TEM)

A transmission electron microscope (JEOL TEM-2100) attached to a CCD camera at a rapid charge of 200 kV was used to determine the size, shape, molecular structure, and morphological data of the acquired CaNPs. Every component of the integrated CaNPs was created by suspending the case on carbon-coated copper networks, allowing allowing dissolvable particles to pass continuously while capturing TEM images (Figure 9). The TEM images were provided by the Central Lab.

### 4.6. CaNP Application Routine

Cucumber fruits were subjected to four different treatments at the same time. The CaNPs treatments included CaNPs-SA 0 mM, CaNPs-SA 1 mM, and CaNPs-SA 2 mM, as well as a control treatment. Fruits were immersed in the CaNPs-SA treatments for 30 min and then stored for 14 days in ambient conditions (20 ± 1 °C, 60% relative humidity).

### 4.7. Physical Quality Determination

Weight loss (WL) percentage was determined as the percentage loss of initial weight at harvest time and was calculated by the following formula: $weight\;loss=\frac{WLi-WLs}{WLi}*100$, where *WL_i_* is the initial weight and *WL_s_* is the weight at a sampling period [48,49].

Cucumber firmness was estimated by utilizing an Instron Universal Testing Machine (Model 4411, Instron, MA, USA) on both sides of the cucumbers in various places on the fruit. A plunger (6 mm diameter) was operated to penetrate the tissues of the skin to the pulp at a depth of 5 mm, and the firmness was measured and presented in N [50].

The cucumber skin color hue was determined using the method described in [51]. The color outline of the fruit was observed with a colorimeter (Minolta CR-300) using the L* a* b* interoperative system proposed by the Commission Internationale de I’ Eclairage (CIE). The assayed color variable (hue angle h°) was verified by applying the equation b*/a*, and the resulting data were represented as recorded in the Minolta camera.

Chlorophyll A and B were determined using N, N-dimethylformamide (DMF) solvent rather than (CH_3_)_2_CO [52]. The extraction was held at 4 °C for 16 h to allow the DMF to eliminate the pigment from the cucumber skin samples [28]. Finally, the extractions were centrifuged for 10 min at 10,000× *g* and measured spectrophotometrically at wavelengths of 663.8 nm (ChA) and 646.8 nm (Ch B); The results were reportedin mg 100 g^−1^ FW.

### 4.8. Cell Wall Enzyme Activities

One gram of cucumber fruit was homogenized with 20 mM Tris-HCl buffer (pH 7). After that, the mixture was centrifuged at 16,000 rpm for 6 min at 4 °C while chilling. Polygalacturonase (PG), cellulase (CEL), and xylanase (XLN) were measured in the filtered sample, which was kept at 20 °C.

In a total volume of 1 mL, we combined polygalacturonic acid and an appropriate amount of enzyme extraction. The addition of the substrate kicked off the process. For one hour, the reaction mixture was maintained at 37 °C. As a result, 500 L of dinitro-salicylic acid reagent was combined and cooked for 10 min in a water bath. The sample was cooled suddenly and the temperature in the lab was attained. With a spectrophotometer, the activities were assessed at 560 nm for PG and XLN and 540 nm for CEL [53].

The lowering of the endpoint of the carboxymethyl cellulose reaction was used to assess cellulase (CEL, EC: 3.2.1.4) enzyme activity [53]. One gram of cucumber was mashed with 20 mM of pH 7.0 Tris-HCL buffer. The mixture was centrifuged for 5 min at 14,000× *g* while being cooled at 4 °C. To consider CEL at 450 nm, the reaction supernatant was stored at 20 °C. As a measure of the chemical, one unit of compound action was transmitted.

The efficiency of pectinase (PT, EC: 3.2.1.15) was determined [54], and the extraction was performed utilizing the technique outlined in [55]. A mixture of 500 L of polygalacturonic acid (0.36 percent *w/v*) with 0.05 M Tris-HCL buffer (pH 8.5), 200 L of 4 mM CaCl_2_, 500 L of chemicals, and 500 L water was used to test the enzyme’s performance. For 3 h, the response mixture was kept at 36 °C. The enzyme was identified by reading the absorption at 232 nm. This method was also used to add up the solvent protein content in the chemical concentrate. The enzyme’s specific activity was measured in milligrams per g^−1^ of protein.

### 4.9. Lipid Peroxidation and Ion Leakage Percentage

A 2.5 g sample of cucumber pulp was mashed in a mortar with 25 mL of 5% (*w/v*) metaphosphoric acid and 500 L of 2% (*w/v*) butylated hydroxytoluene in ethanol and finally homogenized to determine malondialdehyde (MDA) as a terminative of lipid peroxidation [56]. The standard curve was calibrated using various concentrations of 1,1,3,3-tetraethyoxypropane (Sigma) in the range of 0–2 mM (TBARS), which was equivalent to 0–1 mM malondialdehyde (MDA). During the acid warming phase of the test, the 1,1,3,3-tetraethyoxypropane is stoichiometrically converted to MDA [57]. MDA concentrations are used to measure the amount of TBARS present.

Five cucumber pulp discs (5 mm in diameter) were soaked three times in normal water to remove any ions before being placed in 10 mL 0.4 M mannitol alcohol in demineralized water at room temperature for three hours. A conductivity meter was used to check the ion conductivity (EC). All samples were boiled in water for 30 min to kill the cucumber tissue cells before being kept cold in the lab. The EC was then measured again, and the samples’ relative electrolyte leakage was computed using the following equation: ion leakage percentage = [EC after boiling sample—EC after 3-h -/EC after 3-h] × 100 [56].

### 4.10. O_2_^•−^ and H_2_O_2_ Free Radical Production Rate

One gram of cucumber sample was combined with 3 mL of potassium phosphate buffer (50 mM, pH 7.8) under cooling at 4 °C. The reagent was also combined with poly-vinyl-pyrrolidone (PVP 1% *w/v*), then the mixtures were centrifuged at 11,000 rpm at 4 °C for 15 min. The O_2_^•−^ generation rate was measured by examining the NO_2_ age from hydroxylamine in the presence of O_2_^•−^ [45]. A standard bend with NO_2_ was utilized to verify the O_2_^•−^generation rate from the response of O_2_^•−^ to hydroxylamine. The O_2_^•−^product was presented in nmol min^−1^ g^−1^ FW.

For H_2_O_2_ determination, a cucumber sample (1 g) was mixed with 5 mL of 100% (CH_3_)_2_CO then centrifuged at 15,000× *g* for 15 min at 4 °C. One milliliter of the abstraction was mixed with 0.1 mL of 5% Ti (SO4)_2_ and 0.2 mL of NH_4_OH solution. The titanium-peroxide compound was deposited, and the deposit was broken down in 4 mL of 2 M H_2_SO_4_ after centrifugation at 5000× *g* for 25 min; then, it was quantified on a spectrophotometer at 415 nm. The H_2_O_2_ matter was calculated from a standard bend balanced likewise and the fixation presented in ηmol g^−1^ FW [58].

### 4.11. Statistical Analysis

The Software package for co-state Version 6.303 was used to analyze the data (789 lighthouse Ave PMB 320, Monterey, CA 93940, USA) by taking the mean of the two seasons and using a two-way analysis of variance with the calcium nanoparticle/salicylic acid treatments and storage period (days) as factors. Duncan’s multiple range test was used to compare the means of all tested parameters at a *p* < 0.05 level.

## 5. Conclusions

SA was found to reduce the activity of cell wall degradation enzymes, which not only scavenged H_2_O_2_ but also reduced O_2_^•−^ generation in this study. As a result, it is recommended that cucumbers be treated with nano calcium particles mixed with salicylic acid at a concentration of 2 mmol L^−1^ throughout their shelf life.

## Figures and Tables

**Figure 1 molecules-27-03687-f001:**
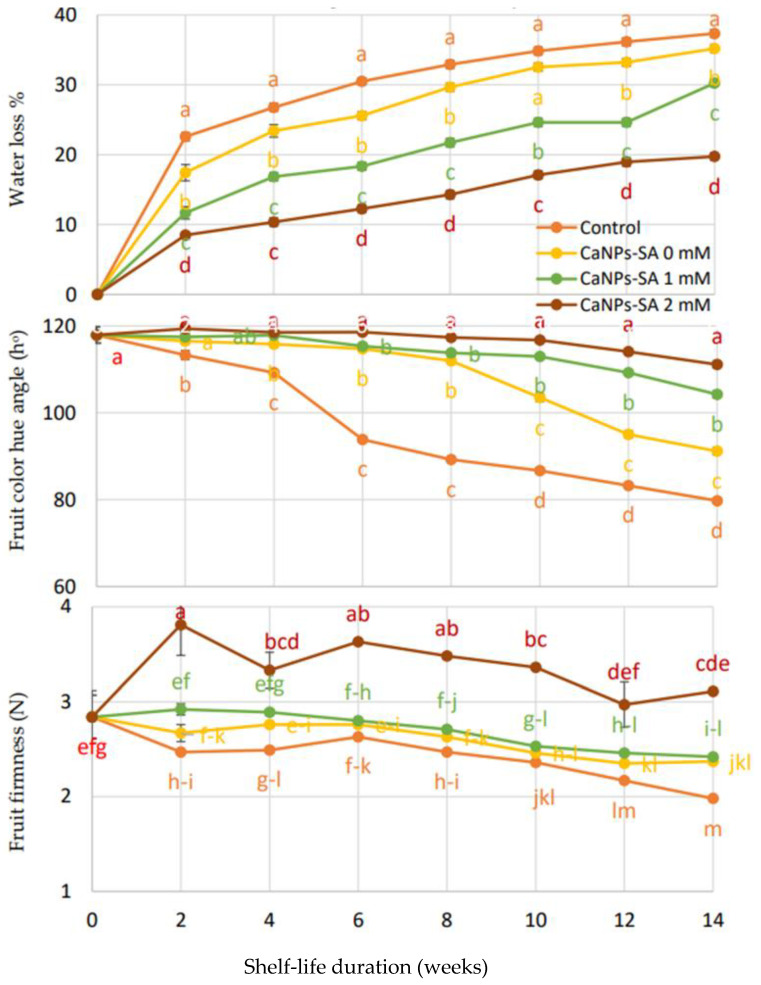
The effects of CaNPs blended with salicylic acid at various doses (0, 1, and 2 mmol L^−1^) on cucumber physical attributes (weight loss percent, fruit color, and firmness) over time. The standard error (*n* = 3) is represented by the vertical bar for the mean of the two seasons.

**Figure 2 molecules-27-03687-f002:**
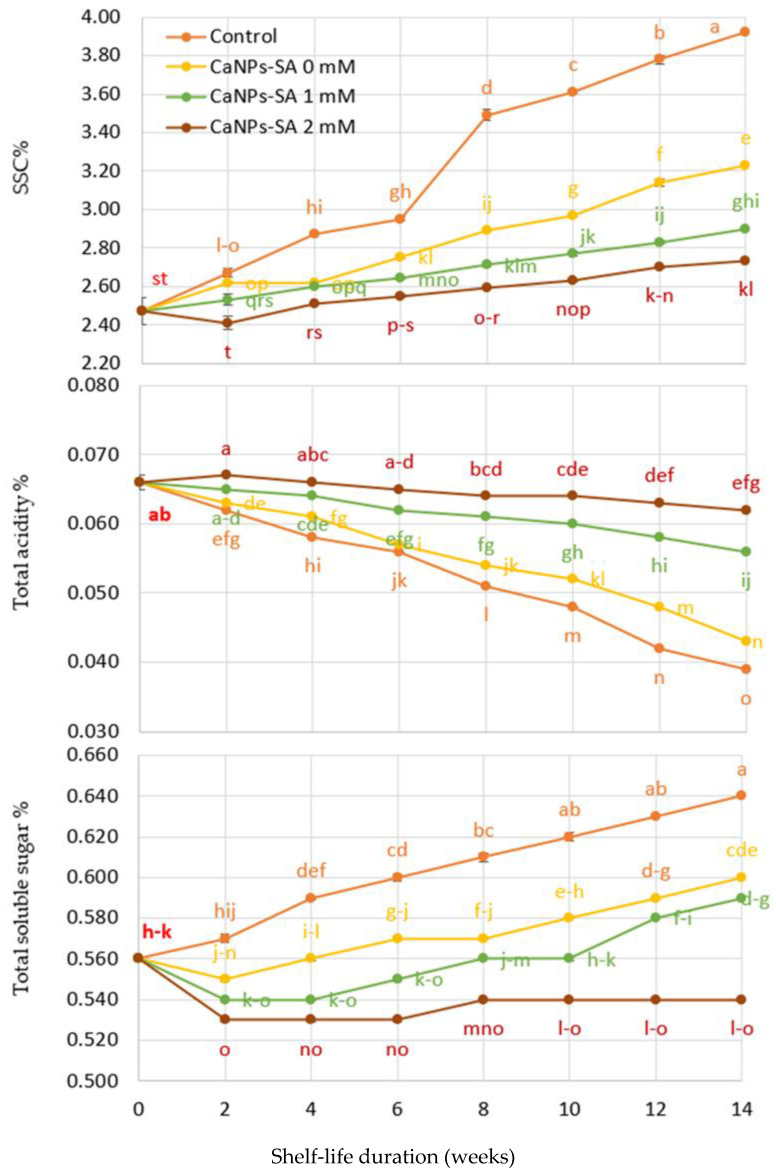
The effects of CaNPs blended with salicylic acid at various doses (0, 1, and 2 mmol L^−1^) on cucumber chemical attributes (SSC%, total acidity %, and total soluble sugar %) over time. The standard error (*n* = 3) is represented by the vertical bar for the mean of the two seasons.

**Figure 3 molecules-27-03687-f003:**
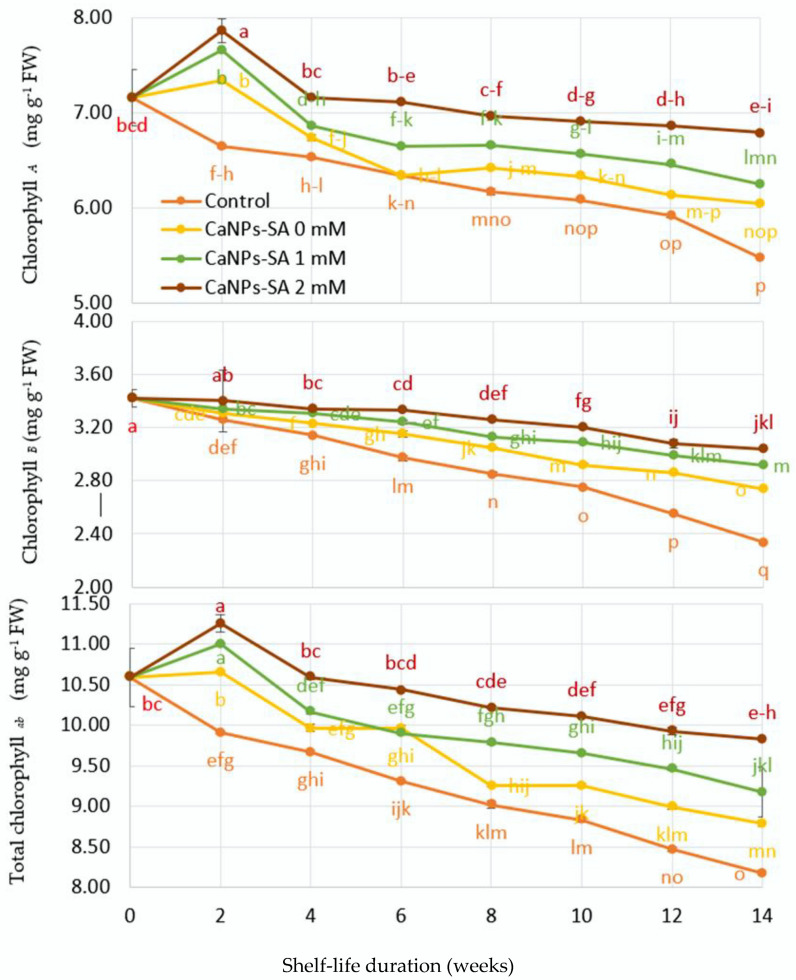
The effects of CaNPs blended with salicylic acid at various doses (0, 1, and 2 mmol L^−1^) on cucumber chemical attributes (chlorophyll compartments) over time. The standard error (*n* = 3) is represented by the vertical bar for the mean of the two seasons.

**Figure 4 molecules-27-03687-f004:**
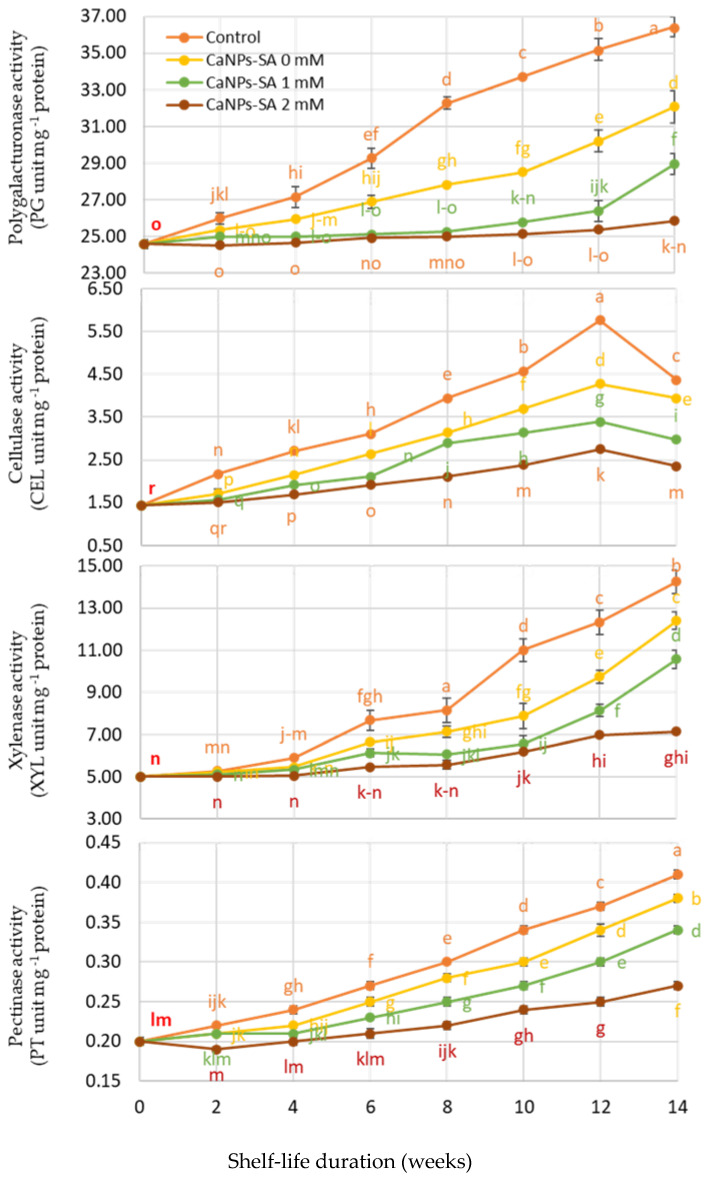
The effects of CaNPs blended with salicylic acid at various doses (0, 1, and 2 mmol L^−1^) on cucumber chemical attributes (cell wall enzyme activities) over time. The standard error (*n* = 3) is represented by the vertical bar for the mean of the two seasons.

**Figure 5 molecules-27-03687-f005:**
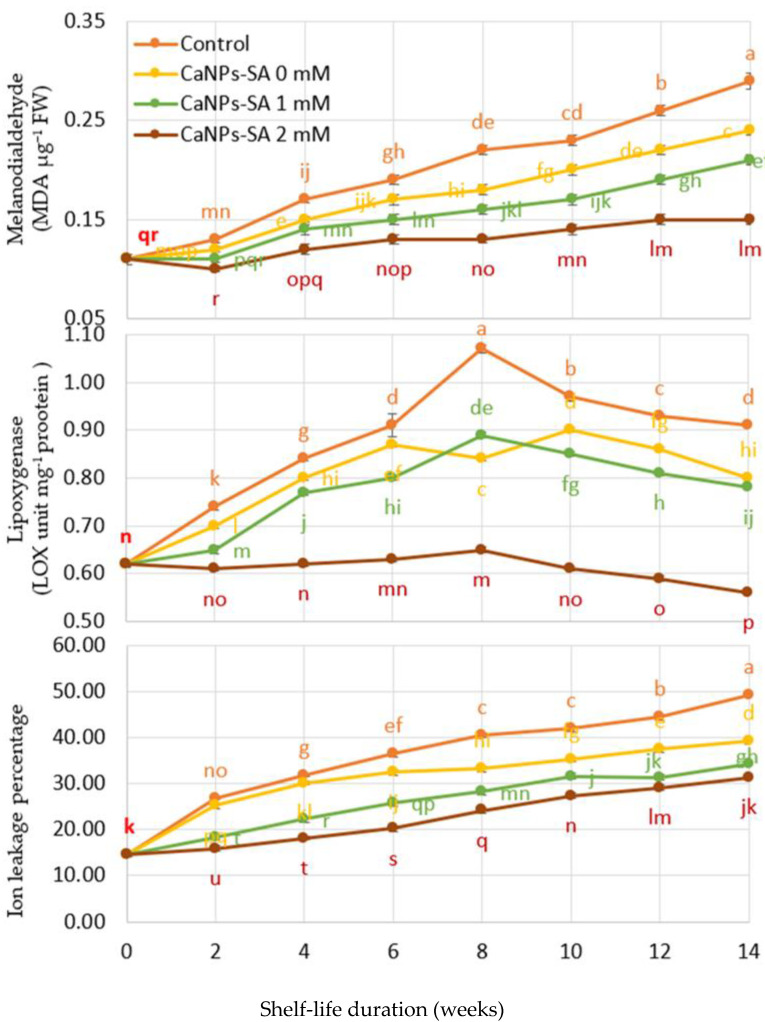
The effects of CaNPs blended with salicylic acid at various doses (0, 1, and 2 mmol L^−1^) on cucumber chemical attributes (MDA, LOX, and ion leakage) over time. The standard error (*n* = 3) is represented by the vertical bar for the mean of the two seasons.

**Figure 6 molecules-27-03687-f006:**
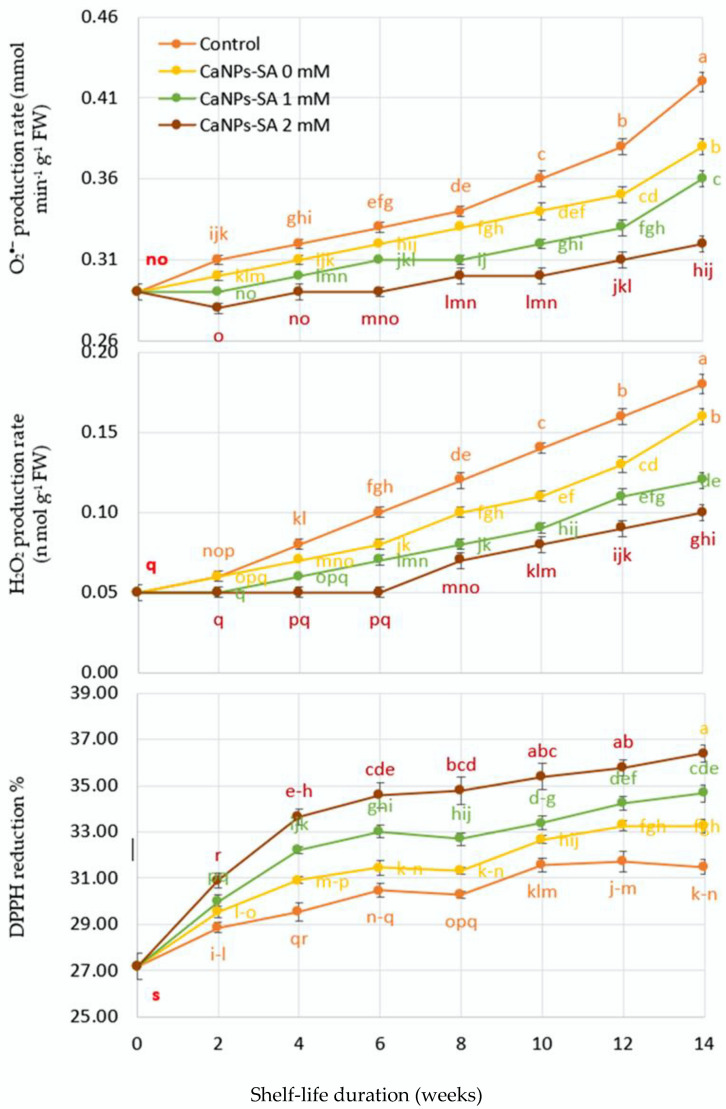
The effects of CaNPs blended with salicylic acid at various doses (0, 1, and 2 mmol L^−1^) on cucumber chemical attributes (O_2_^•−^, H_2_O_2_, and DPPH reduction rate) over time. The standard error (*n* = 3) is represented by the vertical bar for the mean of the two seasons.

**Figure 7 molecules-27-03687-f007:**
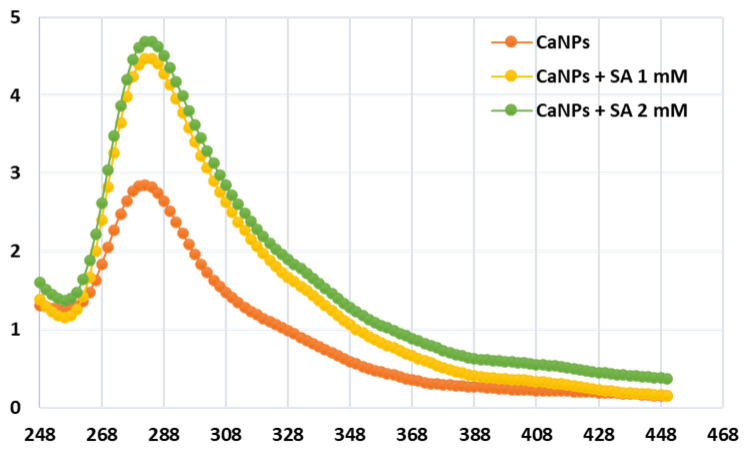
UV-visible absorption spectra of calcium nanoparticles synthesized (CaNPs) blended with various concentrations of salicylic acid (0, 1, and 2 mM) with a peak at 270 nm.

**Figure 8 molecules-27-03687-f008:**
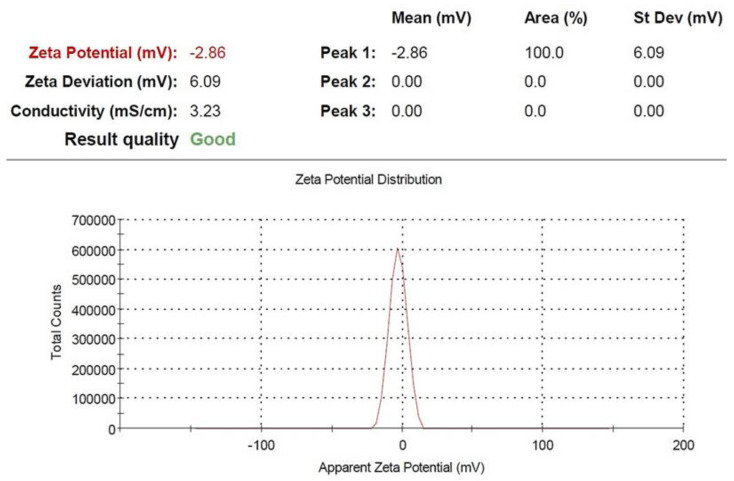
Zeta potential distribution for calcium nanoparticles synthesized with salicylic acid.

**Figure 9 molecules-27-03687-f009:**
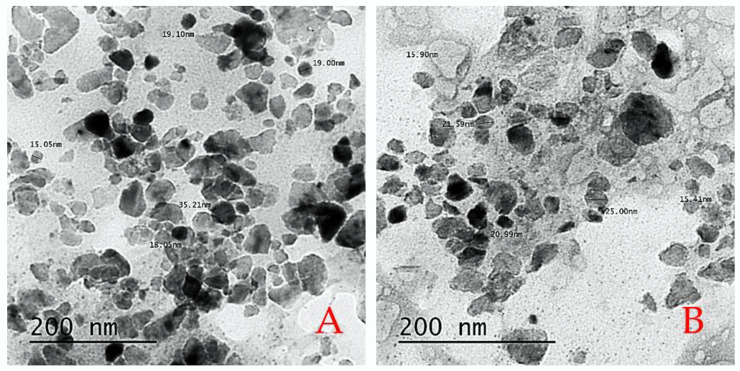
(**A**) The CaNPs had particle sizes ranging from 15.05 to 35.21 nm, while CaNPs containing salicylic acid had particle sizes ranging from 15.90 to 25.00 nm. The particles were mostly spherical, with a few tetragonal particles thrown in for good measure. (**B**) The CaNPs had more aggregated particles than CaNPs containing salicylic acid particles.

## Data Availability

Data is contained within the article.

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
