# Peer review of "The Effect of Combining Post-Harvest Calcium Nanoparticles with a Salicylic Acid Treatment on Cucumber Tissue Breakdown via Enzyme Activity during Shelf Life"

_molecules, 2022, doi:10.3390/molecules27123687_

Round 1

Reviewer 1 Report

The manuscript presents some interesting and significant findings. I have the following observations while going through the manuscript such as

1) The title of manuscript- focus on the cell wall enzyme breakdown but the manuscript has presented other quality parameters, so the title may be changed appropriately as on the postharvest shelf life or quality etc

2) Abstract: The abstract need to rewrite with more clarity with clearly state the design and significant findings. A majority part of the abstract deals with only the parameters studies (lines 29-37), which can be grouped together and briefed. Kindly mention the level of significance should be also included. Similarly, Line 39 is not clear.

3) Keywords: need to add nano calcium particles and salicylic acid

4) Introduction: 

line 47--additionally ----the world. seems too general, please rewrite it

line 63-66: please rephrase

Line 70: high technique may be replaced with advanced technique

5) Materials and Methods: 

Line 94: seasons of 2018 and 2019; please provide the detail of season, average temp, humidity etc 

Line 107: .Some modification. seems incomplete.

Line 108; were were seems repeated

Authors may add justification of the levels selected 1 mM and 2 mM.

Line 146 20 1 should be 20.1 or plus minus 1

Line 168: Minolta camera or chroma meter

Line 176: gramme should be g

Line 233 statistical package for co-statics?

6) Results and Discussion: Appropriate

7) Conclusion: Appropriate 

Author Response

Comments and Suggestions for Authors

The manuscript presents some interesting and significant findings. I have the following observations while going through the manuscript such as

1) The title of manuscript- focus on the cell wall enzyme breakdown but the manuscript has presented other quality parameters, so the title may be changed appropriately as on the postharvest shelf life or quality etc

2) Abstract: The abstract need to rewrite with more clarity with clearly state the design and significant findings. A majority part of the abstract deals with only the parameters studies (lines 29-37), which can be grouped together and briefed. Kindly mention the level of significance should be also included. Similarly, Line 39 is not clear.

Formatted

3) Keywords: need to add nano calcium particles and salicylic acid was added

4) Introduction: 

line 47--additionally ----the world. seems too general, please rewrite it Rephrased

line 63-66: please rephrase  Rephrased

Line 70: high technique may be replaced with advanced technique  Replaced

5) Materials and Methods: 

Line 94: seasons of 2018 and 2019; please provide the detail of season, average temp, humidity etc 

Was added in section 2.6. CaNPs applications regime

Line 107: .Some modification. seems incomplete. Delated

Line 108; were were seems repeated removed

Authors may add justification of the levels selected 1 mM and 2 mM. Those doses are suitable and safe for human

Line 146 20 1 should be 20.1 or plus-minus 1 changed

Line 168: Minolta camera or chroma meter Minolta camera

Line 176: gramme should be g  Replaced

Line 233 statistical package for co-statics? Changed to be correct

6) Results and Discussion: Appropriate Thank you

7) Conclusion: Appropriate Thank you

Reviewer 2 Report

The manuscript aims to investigate the effect of nano-calcium particles treatment on cucumber tissue breakdown enzyme activity during shelf life.

The manuscript should be carefully revised since there are many errors of English. The authors should have also much more care with the data and information. There are many incorrections and typos that need to be revised.

The authors should discuss the reasons to use the concentrations of 1 and 2 mM of salicylic acid. Moreover, the results should be much more discussed to better identify the mechanism that inhibits cell wall degradation enzyme activities, with the blending of salicylic acid and nano particles.

Author Response

The manuscript aims to investigate the effect of nano-calcium particles treatment on cucumber tissue breakdown enzyme activity during shelf life.

The manuscript should be carefully revised since there are many errors of English. The authors should have also much more care with the data and information. There are many incorrections and typos that need to be revised.

All parts of the manuscript were checked in English

 The authors should discuss the reasons to use the concentrations of 1 and 2 mM of salicylic acid. Moreover, the results should be much more discussed to better identify the mechanism that inhibits cell wall degradation enzyme activities, with the blending of salicylic acid and nanoparticles.

Concentrations of salicylic acid were chosen in the safe limits for human nutrition, higher than that is a violation of the recommendations of the European Union, for example, in the circulation of horticultural crops between countries
The role of salicylic acid in inhibiting the activity of root-degrading enzymes in maintaining the quality of cucumber fruits during trading was discussed.

Round 2

Reviewer 2 Report

The manuscript continues with errors in English and in the presentation of the data. I send in the attached file some incorrections that need to be changed. However, I was not exhaustive in this aspect.

The authors should improve the English of the manuscript since some aspects are unclear.

In section 2.11 the authors refer that it was done the mean of the two seasons. Thus, the data presented in the figures correspond to the mean value of the parameter of the two seasons?  And the n=3 (referred in figures)?

Author Response

The authors should improve the English of the manuscript since some aspects are unclear.

All manuscript parts were checked in English and grammar

In section 2.11 the authors refer that it was done the mean of the two seasons. Thus, the data presented in the figures correspond to the mean value of the parameter of the two seasons?  And the n=3 (referred in figures)?

All figurs were fixed 

This manuscript is a resubmission of an earlier submission. The following is a list of the peer review reports and author responses from that submission.